# Risk Factors for Falls and Fall-Related Fractures in Community-Living Older People with Pain: A Prospective Cohort Study

**DOI:** 10.3390/ijerph20116040

**Published:** 2023-06-02

**Authors:** Tatsuya Hirase, Yoshiro Okubo, Kim Delbaere, Jasmine C. Menant, Stephen R. Lord, Daina L. Sturnieks

**Affiliations:** 1Division of Physical Therapy Science, Graduate Course of Health and Social Work, Kanagawa University of Human Services, Kanagawa 238-8522, Japan; 2Falls, Balance and Injury Research Centre, Neuroscience Research Australia, Sydney, NSW 2031, Australia; 3Faculty of Medicine and Health, University of New South Wales, Sydney, NSW 2031, Australia

**Keywords:** accidental falls, pain, aged, balance, gait, mobility, physical activity

## Abstract

(1) Background: This prospective study aimed to identify predictors of falls and fall-related fractures in community-dwelling older people with pain; (2) Methods: Participants comprised 389 community-dwelling older people aged 70+ years who had musculoskeletal pain in the neck, back, hip, leg/knee and/or feet. Demographic, anthropometric, balance, mobility, cognitive function, psychological status and physical activity level measures were obtained at baseline. Falls were monitored with monthly falls calendars for 12 months. Logistic regression analyses were performed to identify predictors of falls and fall-related fractures during a 12-month follow-up; (3) Results: Of the 389 participants, 175 (45.0%) and 20 (5.1%) reported falls and fall-related fractures during the 12-month follow-up, respectively. Greater postural sway on foam, more depressive symptoms and lower physical activity levels at baseline were associated with falls during the 12-month follow-up. Slower walking speed at baseline was associated with fall-related fractures during the 12-month follow-up. These associations remained significant after adjusting for age, sex, body mass index, comorbidities and medication use; (4) Conclusions: This study suggests poor balance, low mood and a less active lifestyle are predictors of falls, and slower walking speed predicts fall-related fractures among community-dwelling older people with pain.

## 1. Introduction

One in three people aged 65 years or older fall at least once annually and about half fall recurrently [1,2]. Between 20% and 30% of older people who experienced a fall suffer moderate to severe injuries such as hip and wrist fractures and head trauma, which can lead to disability, nursing home admission and mortality [3,4]. Fatal and non-fatal fall-related injuries such as bone fractures in older people are associated with substantial medical costs totaling $50 billion in 2015 in the United States [4,5].

Pain is prevalent in older people living in the community with reported rates of up to 53% [6,7]. Due to population ageing, the number of older people with pain and its population-wide burden are projected to increase markedly in the next 50 years [8]. Pain has been identified as a significant risk factor for falls in older people, and systematic review evidence suggests that those with pain have a two-fold increased risk of falling compared to those without pain [9,10]. However, studies on risk factors for falls have focused on older people primarily without pain [11,12] with little information available on the fall-risk profile of older people with pain to inform possible intervention strategies.

Previous studies have reported that pain has a negative impact on physical function, including muscle strength, balance, mobility and reaction time [13]. A systematic review has indicated that pain in community-dwelling older people can impair aspects of balance control during standing, daily-life movements, and when reacting to a postural perturbation [14]. In addition, there is evidence suggesting that pain may be associated with poorer cognitive function measures, such as executive function, as well as a reduced psychological status, such as a concern about falling and depressed mood [13,15,16]. In addition, a systematic review has indicated that older people with pain have significantly lower physical activity levels compared to those without pain [17]. These findings suggest that physical, cognitive, psychological and lifestyle factors may contribute to an increased risk of falls in older people living with pain. However, no studies have investigated whether the above factors predispose older people with pain to falls and fall-related fractures. As this information would be valuable for informing strategies for preventing falls, we conducted a prospective study involving a comprehensive range of explanatory predictor variables to identify the most salient factors that predict falls and fall-related fractures in older people with pain. We hypothesized that multiple physical, psychological and lifestyle factors would be independently associated with falls in older people with pain.

## 2. Materials and Methods

### 2.1. Participants and Study Design

This 12-month prospective study involved an analysis of merged data collected at the Falls, Balance and Injury Research Centre at Neuroscience Research Australia, in association with the Sydney Memory and Ageing Study, which was conducted between 2004 and 2010 (Waves 1 to 4) [18]. Using the electoral roll for Eastern Sydney, community-dwelling people aged 70 years or over were randomly sampled and invited to participate in the study. Inclusion criteria were aged 70+ years, independent community-living, and able to walk 400 m without assistance. Furthermore, this study focused on those who reported having musculoskeletal pain in the feet, knee/leg, hip, back and/or neck [14]. Pain was assessed by the yes/no question “Do you currently suffer from any of the following: pain in the neck/back, hip, knee/leg and/or feet?”. Older people who responded “yes” to this question were included. Exclusion criteria for this study were cardiovascular, musculoskeletal or neurological, impairments including motor neuron disease, multiple sclerosis, dementia, central nervous system inflammation, psychotic symptoms, and psychological or medical conditions, which could interfere with assessments, minimal English language skills, and a score of <24 on the Mini-Mental State Examination [19]. The study protocol was reviewed and approved by the University of New South Wales Human Research Ethics Committee. Written informed consent was obtained from all participants prior to participation.

### 2.2. Assessments

#### 2.2.1. Falls

A fall was defined as “an unexpected event in which the person comes to rest on the ground, floor, or lower level” [20]. Participants were prospectively followed-up for falls and fall-related fractures for 12 months using monthly falls calendars. Participants were instructed to return a completed falls calendar to research staff each month via postal mail. If a calendar failed to be returned within two weeks, research staff contacted the participants via telephone to obtain the falls data. Fallers were defined as those who reported one or more falls during the 12-month follow-up period.

#### 2.2.2. Demographics

At baseline, participant age, sex, anthropometrics, comorbidities and medication use were assessed via questionnaires and clinical assessments. Body mass index (BMI) was calculated by dividing the body weight (kg) by height squared (m^2^). We defined comorbidity as having at least two of the following major medical conditions; diabetes mellitus, stroke, Parkinson’s disease, heart disease and chronic obstructive pulmonary disease. Taking four or more medications was classified as polypharmacy.

#### 2.2.3. Physical Function

At baseline, participants underwent physical function assessments involving tests of lower limb proprioception [21], knee extension muscle strength [21], simple hand-reaction time [21], standing postural sway [21], sit-to-stand performance (STS) [22], timed up and go (TUG) [23], and 6 m walking time [24]. Proprioception was measured as the average alignment errors (deg) from five trials of a task requiring aligning the great toes either side of a vertical protractor while seated and with eyes closed. Lower limb muscle strength was measured in the dominant leg using a custom-built strain gauge. Participants were seated on a chair with knee flexed to 90 deg and the strain gauge was attached to the lower leg. The maximal isometric knee extension force (kg) value from three trials was used. Hand reaction time was measured using a light stimulus device and a mouse. The time from presentation of the light stimulus to a finger pressing the mouse was recorded, and the average of 10 trials was used. Postural sway was measured while participants stood with their feet hip-width apart on a foam rubber pad (70 cm × 60 cm × 15 cm thick) with eyes open for 30 s, using a sway meter that recorded displacements (millimeters) of the body at the level of the waist. The device consisted of an inflexible 40-cm-long rod with a vertically mounted pen at its end. The rod was mounted on a 20 cm wide metal plate which was fitted over the participant’s lower back (level of the posterior superior iliac spine) by a firm belt so that the rod extended posteriorly. Fitted firmly, the sway meter offers 1 degree of freedom between the belt and pen, as it is free to move in the pitch plane. The pen recorded participant’s postural sway on a sheet of millimeter graph paper, fastened to the top of an adjustable-height table. STS was evaluated as the time (seconds) taken to rise from a chair five times as fast as possible with the participants’ arms folded across their chest. TUG was measured as the time (s) taken to stand up from a standard chair, walk 3 m as fast as possible, turn, walk back to the chair and sit down again. We also conducted a six-meter walk test on a 10 m path in a corridor. Participants were instructed to walk at their usual, comfortable walking pace. The time taken to walk the middle 6 m of a 10 m path was recorded in seconds.

#### 2.2.4. Cognitive Function and Psychological Status

At baseline, trained Research Assistants assessed cognitive function and psychological status. Executive function was assessed using the Trail Making Test (TMT) and the difference between parts A and B of the test was calculated to provide a measure of executive function [25]. In addition, processing speed was assessed using the Digit Symbol Substitution Test (DSST) [26]. Depressive symptoms and concern about falling as markers of psychological status were assessed using the 15 item Geriatric Depression Scale (GDS-15) [27] and the Falls Efficacy Scale-International (FES-I) [28], respectively.

#### 2.2.5. Physical Activity

Participant baseline levels of physical activity was evaluated using the Incidental and Planned Exercise Questionnaire (IPEQ) validated for older people [29]. The IPEQ estimates the average physical activity time (hours per week) over the past three months. These include planned physical exercise, in addition to planned walks (e.g., walking in the park) and incidental physical activities (e.g., walking to the store, gardening). The planned physical exercise includes participation in exercise classes, home exercise, other sport/exercise, such as swimming, golf, bowls, dancing and jogging. The time spent undertaking all panned and incidental activities is summed to provide an estimate of total physical activity levels (h/week).

### 2.3. Statistical Analysis

All data were checked for normal distribution using skewness statistics (within −1 and 1) and log transformed if needed. The participants with and without falls were compared. Student *t*-tests and X^2^ tests were used to examine between-group differences in continuous and categorical variables, respectively. We also used these tests to examine differences between those with and without fall-related fractures. Variables that were found with significant between-group differences in the above univariable tests were entered as independent variables in separate, logistic regression analyses with falls (yes/no) and fall-related fractures (yes/no) as dependent variables. The independent variables were converted to Z-scores based on the means and standard deviations. Initial models (Models 1) were unadjusted and Models 2 were adjusted for age, sex, BMI, comorbidity and polypharmacy. Finally, the independent and significant predictor variables for the fall outcome in Model 2 were dichotomized using the Youden index, summed and contrasted against the occurrence of falls in the follow-up period. All analyses were conducted using SPSS 25.0 for Windows (SPSS Inc., Armonk, NY, USA). *p* < 0.05 was considered statistically significant.

## 3. Results

### 3.1. Baseline Participant Characteristics

A total of 814 Sydney Memory and Ageing Study participants contributed to the falls and balance sub-study across waves one to four. Of these 814 participants, 389 (212 women, 177 men, average age 78.9 years, standard deviation 4.8 years) reported having bodily pain, completed the baseline clinical assessments and a 12-month follow-up for falls, and were therefore included in these analyses (Figure 1).

Table 1 shows baseline participant characteristics and test results for those with and without falls, and those with and without fall-related fractures during the 12-month follow-up. Of the 389 participants, 175 (45.0%) reported falls and 20 (5.1%) reported fall-related fractures during 12-months follow-up.

Compared to non-fallers, fallers had higher baseline BMI (*p* = 0.040), increased postural sway (*p* = 0.011), higher FES-I scores (*p* = 0.019), higher GDS-15 scores (*p* = 0.034), and lower total physical activity levels (*p* = 0.018).

The participants with fall-related fractures had increased postural sway (*p* = 0.017), slower 6 m walking time (*p* = 0.012) and were shorter in stature (*p* = 0.043), compared to those without fall-related fractures.

### 3.2. Logistic Regression Analyses to Determine Predictors of Falls and Fall-Related Fractures

Results of the unadjusted (Model 1) and adjusted (Model 2) models are presented in Table 2. In Model 1, increased postural sway, higher GDS-15 and FES-I scores, and lower IPEQ total activity were significantly associated with falls. Of these variables, increased postural sway, higher GDS-15 scores and lower IPEQ total activity remained significantly associated with falls when adjusting for age, sex, BMI, comorbidity and polypharmacy (Model 2).

Regarding fall-related fractures (Table 3), increased postural sway and slower 6 m walking times were associated in the unadjusted model (Model 1). When adjusted for age, sex, BMI, comorbidity and polypharmacy, slower 6 m walking times remained significantly associated with fall-related fractures (Model 2).

### 3.3. Proportion of Fallers in Relation to Number of Risk Factors

The cut points for risk factors identified using the Youden index were ≥200 mm for postural sway, ≥3 for GDS and ≤34 h per week for IPEQ total activity. The proportions of participants who experienced falls in those with zero, one, two and three risk factors were 25.7%, 41.3%, 57.1% and 62.2%, respectively (Figure 2).

## 4. Discussion

This longitudinal study comprehensively examined factors that predict falls and fall-related fractures in community-dwelling older people with pain. Our analyses revealed that poorer balance in a challenging condition (postural instability while standing on a foam rubber pad), depressive symptoms and physical inactivity were associated with prospective falls independently from age, gender, BMI, comorbidity and polypharmacy. In addition, slower gait speed predicted fall-related fractures independently from the above covariates. Consistent with our hypothesis that multiple physical, psychological and lifestyle factors would be associated with falls, we found poorer balance, low mood and physical inactivity were independent risk factors for falls in older people with pain.

The current finding suggesting that poorer balance contributes to falls in older people with pain is in line with systematic review evidence reporting the association between pain and balance impairment in older people [14]. Pain may contribute to poor balance via several mechanisms. Pain disrupts proprioceptive processing between muscle, joints, and cortical systems [30], and this interference could result in increased body sway, as somatosensory inputs (joint, muscle, and skin mechanoreceptors) are required for the constant monitoring of the center of mass position [31,32]. Moreover, it has been reported that pain is associated with trunk or lower limb muscle weakness [33,34] which could then lead to postural instability and falls in older people [35,36,37].

Our results indicated that depressive symptoms and physical inactivity are fall risk factors in older people with pain. These observations agree with previous studies suggesting inter-relationships between pain and physical, psychological and lifestyle risk factors for falls [2,11]. For example, pain in community-dwelling older people has previously been shown to be associated with poor balance [14], depressive symptoms [15], slow walking speed [6,11] and low physical activity levels [17]. Poor balance can also restrict physical activity [38], which can in turn lead to depressive mood [39]. Impaired reactive balance has been shown to partially mediate the association between neck/back pain and fall-related fractures in older people [40]. Moreover, it has been reported that pain-related fear leads to avoidance behaviors, and hypervigilance to bodily sensations followed by disuse and disability in people with pain [41], which can further increase the risk of falling [42]. In this cohort with reported pain, it seems the related issues of poor balance, depressive symptoms and inactivity added cumulatively to fall risk. Indeed, the proportion of participants who experienced falls increased from 25.7% in those with no risk factors to 62.2% in those with all three risk factors.

The finding that slow gait speed was the strongest risk factor for fall-related fractures may indicate this measure encapsulates both fall risk due reduced physical functioning, as well as poor health and frailty [43]. In addition to several demographic, anthropometric, lifestyle and body composition factors [44,45], some cross-sectional [46,47] and longitudinal [48] studies have shown that slow gait speed is also significantly associated with reduced bone strength. This may partially explain why slow gait in older people with pain presented an increased risk of fall-related fractures.

The cut points for the three independent risk factors for predicting falls using the Youden index requires discussion. The best cut point for postural sway while standing on a foam rubber pad was 200 mm. This value is higher than that reported in studies of community-dwelling older people where the best cut points for predicting falls have ranged from 105 to 168 mm [49,50,51]. It is likely that the higher cut-point found here reflects an overall worse balance in our sample with pain [14]. Our finding that the relatively low score of three on the GDS-15, best discriminated between fallers and non-fallers, suggests that even mild depressive symptoms is a risk factor for falls in older people with pain [52]. Finally, to our knowledge, this is the first study to investigate a cut-point of combined incidental and planned physical activities to predict falls in older people with pain. Our cut-point indicating less than 34 h/week of total physical activity is predictive of future falls, and is consistent with previous prospective cohort studies reporting that physical inactivity increases fall risk [53,54]. However, there is also evidence of non-linear (e.g., u-shaped) associations between falls and physical activity [42] with a greater fall risk in inactive/impaired older people, as well as active older people with high environmental exposure. Thus, cut points may vary depending on sample characteristics, such as frailty and disability.

Our findings provide important information that may be useful in the design of effective fall preventive strategies. It is likely that addressing balance, mood and activity levels, as well as pain management would be an effective fall prevention strategy for community-dwelling older people with pain. Encouragingly, systematic review evidence suggests that exercise programs are effective in addressing poor balance, low mood and inactivity [55,56], as well as in preventing falls in community-dwelling older people [57,58], and that exercise programs that challenge balance and have a high dose (i.e., 3 or more hours per week of prescribed exercise) have the largest effects for fall prevention in older people [57]. Previous studies have also reported that strength and balance training programs significantly improve walking speed in community-dwelling older people with pain [59,60]. Thus, it is likely that moderate–high intensity balance training would also be an efficacious fall prevention strategy for those with pain. Moderate–high intensity balance training is defined as exercise while standing and moving the center of mass, narrowing the base of support, and minimizing upper limb support [57]. For example, our systematic review and meta-analysis has demonstrated that both reactive and volitional stepping interventions can prevent falls by approximately 50% in older people in community and institutional settings [61], and may be particularly beneficial in this regard.

Strengths of the study include the representative large sample of community-dwelling older people with pain and the comprehensive assessments of potential predictor variables. However, there are certain study limitations which should be acknowledged. Firstly, we did not assess participant characteristics including social life and employment that can influence depression [62], and previous surgery on the spine or joints that can influence balance and strength [63,64]. In addition, we queried participants about their pain only in the feet, knee/leg, hip, back and/or neck. Thus, pain intensity, pain duration, number of pain sites and use of analgesics were not included in our analyses. Therefore, these findings may not be generalizable to those with musculoskeletal pain in other body regions, chronic widespread pain, neuropathic pain and pain related to conditions such as fibromyalgia and arthritis that requires surgery. Secondly, our study participants had relatively high physical function and good mental health, so our findings may not be generalizable to people with physical or neurological impairments or physical frailty. Thirdly, we did not examine the impact of social isolation and loneliness on falls. Finally, the causal relationship between pain and physical, cognitive, psychological and lifestyle risk factors for falls are difficult to tease out because many of these factors are chronic and some may have existed before the commencement of this prospective study. Further research detailing participant characteristics, pain assessments and social factors, and a longer-term follow-up is required to confirm the current findings.

## 5. Conclusions

This study suggests poorer balance, low mood and physical inactivity are risk factors for falls, and that slower walking speed is a risk factor for fall-related fractures among older people with pain. Strategies to improve balance, mobility and mental health, promote an active lifestyle, as well as addressing pain directly, may facilitate fall prevention interventions for community-dwelling older people with pain. Further prospective cohort studies including detailed participant characteristics, pain assessments and social factors are required to confirm the current findings.

## Figures and Tables

**Figure 1 ijerph-20-06040-f001:**
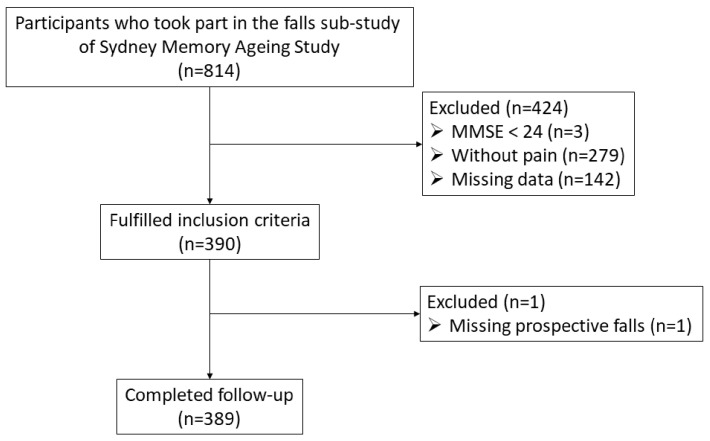
Flow chart of participants included in the study.

**Figure 2 ijerph-20-06040-f002:**
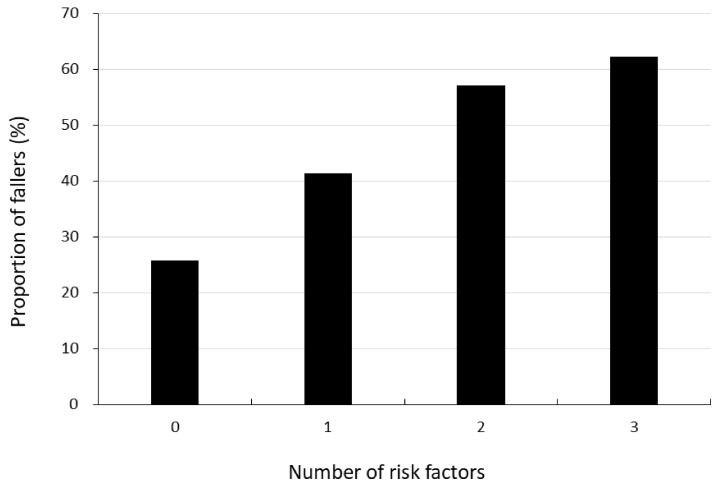
Absolute risk of falls in relation to number of risk factors. Risk factors for falls identified from the logistic regression analysis: Postural sway on a foam rubber pad ≥ 200 mm/30 s, GDS-15 score ≥ 3 point and IPEQ ≤ 34 h/week.

**Table 1 ijerph-20-06040-t001:** Baseline characteristics of participants with and without prospective falls and fall-related fractures.

	12-Month Follow-Up					
	Falls			Fall-Related Fractures		
	Yes (*n* = 175)	No (*n* = 214)	*p*-Value	Yes (*n* = 20)	No (*n* = 369)	*p*-Value
Age (years)	79 (4.8)	78.8 (4.8)	0.638	81 (6.4)	78.8 (4.7)	0.132
Female, *n* (%)	98 (56.0)	114 (53.3)	0.591	14 (70.0)	198 (53.7)	0.153
Height (cm)	163.7 (9.3)	164.0 (9.5)	0.772	159.8 (8.7)	164.1 (9.4)	0.043
Weight (kg)	75.6 (15.8)	73.3 (14.2)	0.129	69.0 (11.5)	74.7 (15.1)	0.100
BMI (kg/m^2^)	28.2 (5.3)	27.1 (4.1)	0.040	27.0 (3.4)	27.6 (4.7)	0.540
Comorbidity (yes)	11 (6.3)	11 (5.1)	0.658	2 (10.0)	20 (5.4)	0.401
Stroke	1 (0.6)	9 (4.2)	0.024	1 (5.0)	9 (2.4)	0.483
Parkinson’s disease	1 (0.6)	1 (0.6)	0.886	0 (0)	2 (0.5)	0.741
Diatebes mellitus (type 1 or type 2)	26 (14.9)	23 (10.8)	0.223	4 (20.0)	45 (12.3)	0.311
Chronic obstructive pulmonary disease	3 (1.7)	2 (0.9)	0.504	0 (0)	5 (1.4)	0.599
Heart disease	41 (23.4)	47 (22.1)	0.750	4 (20.0)	84 (22.8)	0.769
Use of 4 or more medications (yes)	110 (62.9)	149 (69.6)	0.183	13 (65.0)	246 (66.7)	0.874
Physical function						
Proprioception (deg)	2.3 (1.4)	2.4 (1.5)	0.842	2.5 (1.2)	2.3 (1.5)	0.419
Lower limb muscle strength (kg)	27.5 (11.2)	28.4 (12.9)	0.464	25.7 (11.6)	28.1 (12.2)	0.267
Hand reaction time (ms)	235 (37.5)	237 (47.8)	0.932	245 (31.7)	236 (43.9)	0.126
Postural sway on a foam rubber pad (mm/30 s)	214 (117.0)	184 (107.0)	0.011	258 (143.8)	194 (109.9)	0.017
STS (s)	15.8 (4.5)	16.0 (5.8)	0.735	17.4 (5.8)	15.9 (5.2)	0.237
TUG (s)	9.9 (3.6)	10.1 (3.9)	0.586	11.2 (4.0)	9.9 (3.8)	0.174
6 m walking time (s)	9.4 (3.5)	9.4 (3.3)	0.516	12.3 (5.5)	9.3 (3.2)	0.012
Cognitive function						
TMT B-A (s)	72.5 (47.4)	71.5 (45.7)	0.842	83.8 (60.1)	71.4 (45.7)	0.284
DSST (score)	49.0 (11.3)	47.9 (12.4)	0.381	44.6 (13.6)	48.6 (11.9)	0.153
Psychological status						
GDS-15 (score)	2.6 (2.2)	2.2 (1.9)	0.034	2.8 (2.5)	2.3 (2.0)	0.333
FES-I (score)	24.6 (7.3)	23.1 (7.1)	0.019	26.9 (7.8)	23.6 (7.2)	0.077
Physical activity						
IPEQ-total (hours/week)	29.9 (15.1)	32.7 (16.3)	0.018	28.6 (16.3)	30.7 (15.9)	0.634

Data are presented as mean (standard deviation) or number (percentage). Comorbidity indicates the presence of 2 or more of: diabetes mellitus, chronic obstructive pulmonary disease, heart disease, stroke, and Parkinson disease. BMI: Body Mass Index. STS: Sit-to-Stand performance. TUG: Timed Up and Go test. TMT: Trail Making Test. DSST: Digit Symbol Substitution Test. GDS-15: 15-item Geriatric Depression Scale. FES-I: Falls Efficacy Scale-International. IPEQ: Incidental and Planned Exercise Questionnaire.

**Table 2 ijerph-20-06040-t002:** Predictors of any falls during 12-month follow-up.

	Dependent Variables Any Falls					
	Model 1			Model 2		
	OR	95% CI	*p*-Value	OR	95% CI	*p*-Value
Postural sway on a foam rubber pad	1.29	1.06–1.57	0.012	1.38	1.10–1.73	0.005
GDS-15	1.23	1.01–1.50	0.037	1.26	1.02–1.55	0.034
FES-I	1.21	1.00–1.46	0.041	1.18	0.96–1.45	0.109
IPEQ-total	0.72	0.58–0.90	0.004	0.72	0.57–0.91	0.006

Model 1: unadjusted, univariable models. Model 2: multivariable model adjusted for age, sex, BMI, comorbidity and polypharmacy (use of 4 or more medications). GDS-15: 15-item Geriatric Depression Scale. FES-I: Falls Efficacy Scale-International. IPEQ: Incidental and Planned Exercise Questionnaire. OR: Odds Ratio. CI: Confidence Interval.

**Table 3 ijerph-20-06040-t003:** Predictors of fall-related fractures during 12-month follow-up.

	Dependent Variable: At Least One Fall with Fracture					
	Model 1			Model 2		
	OR	95% CI	*p*-Value	OR	95% CI	*p*-Value
Postural sway on a foam rubber pad	1.41	1.05–1.90	0.024	1.40	0.99–1.97	0.059
6 m walking time	1.67	1.24–2.23	0.001	1.58	1.14–2.19	0.006

Model 1: unadjusted, univariable models. Model 2: multivariable model adjusted for age, sex, BMI, comorbidity and polypharmacy (the use of 4+ medications). OR: Odds Ratio. CI: Confidence Interval.

## Data Availability

Data sharing is not applicable to this study.

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
