# Peer review of "Risk Factors for Falls and Fall-Related Fractures in Community-Living Older People with Pain: A Prospective Cohort Study"

_ijerph, 2023, doi:10.3390/ijerph20116040_

Round 1

Reviewer 1 Report

Thank you for your manuscript. Overall is well explained and presented. There are, however, some minor corrections need for improving the MS.

1. Method; physical function- Please indicate the model of the tested equipment for repeatability in the further research. For example, a sway meter. and detail of foam height uses

2. Please explain why the neck pain area was included for those suffering with pain in this study. Have you had the information of the pain area for those falls participant?

3. Limitation; This study is well organized and investigated in multidimensional of health-related  which includes physical and psychological issues. However, social issue can also be a factor of falls. The further study should also examine on this dimension as well.

Author Response

Responses to the comments from Reviewer #1:

Thank you for your manuscript. Overall is well explained and presented. There are, however, some minor corrections need for improving the MS.

  1. Method; physical function- Please indicate the model of the tested equipment for repeatability in the further research. For example, a sway meter. and detail of foam height uses

Response: We have now indicated the detail of the assessments of postural sway as follows:

“Postural sway was measured using a sway meter that recorded displacements of the body at the level of the waist in millimeters while participants stood on a foam rubber pad (70 cm × 60 cm × 15 cm thick) with eyes open for 30 seconds. The device consisted of an inflexible 40-cm-long rod with a vertically mounted pen at its end. The rod was mounted on a 20 cm wide metal plate which was fitted over the participant’s lower back (level of the posterior superior iliac spine) by a firm belt so that the rod extended posteriorly. Fitted firmly, the sway meter offers 1 degree of freedom between the belt and pen as it is free to move in the pitch plane. The pen recorded participant’s postural sway on a sheet of millimeter graph paper, fastened to the top of an adjustable-height table.” (page 3, lines 110-119)

  1. Please explain why the neck pain area was included for those suffering with pain in this study. Have you had the information of the pain area for those falls participant?

Response: Thank you for this comment. Our previous systematic review evidence suggests that neck, back, hip, knee and feet pain in older adults is associated with poor balance identified as a main risk factor for falls (Hirase T, et al. J Am Med Dir Assoc 21: 597-603, 2020). Thus, we included the neck pain area in this study and have provided a reference in the manuscript to support this methodological approach.

“Further, this study focused on those reporting having musculoskeletal pain in the neck/back, hip, knee/leg and/or feet [14].” (page 2, lines 71-72)

  1. Limitation; This study is well organized and investigated in multidimensional of health-related which includes physical and psychological issues. However, social issue can also be a factor of falls. The further study should also examine on this dimension as well.

Response: Based on your suggestion, we have now stated the need of the further study that investigates whether social issues predispose older people with pain to falls and fall-related fractures in the limitation sections as follows:

“Third, we did not examine the impact of social isolation and loneliness on falls.” (page 9, lines 301-302)

“Further prospective cohort studies including detailed participant’s characteristics, pain assessments and social factors are required to confirm the current findings.” (page 9, lines 316-318)

Reviewer 2 Report

Title of paper           

Risk factors for falls and fall-related fractures in community-dwelling older people with pain: a prospective cohort study

General comments

This is a topical issue and the authors deserve appreciation for their work aimed at elucidating associations between physical functional characteristics, cognitive function, psychological status and fall-related fractures in older population with complaints of pain.

The description of participants and applied methods is sufficiently clear and the number of participants (389) allows for opening the chosen topics.

The statistical analysis allows the comparison between groups and revealing associations between the measured characteristics. The collected data is presented in three tables and one figure.

In the Discussion the authors unnecessarily repeat the literature review, instead, more comparisons between the collected data and the results from previous publications should be added.

The list of references provides an overview of the related studies, however, the share of the article’s authors in the citations is 16% (n=8 from the total of 50). It is recommended to include more references dating from recent years.

Specific comments

The title of the manuscript reflects well the main idea and adequately describes the participants of the conducted research.

Abstract and key words

They are written in the logical manner and include all needed components.

Introduction

Please add the hypothesis of the study.

Materials and Methods

Participants

Please add the following data:

- previous and/or current sport/fitness activity of participants

- social life (possible influence on the depression level) – family status, partners, children

- employment (elderly persons may continue working)

- previous surgery on the spine or joints (replacements, etc.) that can influence body balance and strength

If the above components were not estimated, include them in the limitations of the study.

Physical function estimation:

L 107 – add the dimensions of foam pad (width x length x height) and the producer.

Discussion

Recommendations:

- First present the results of your own study and then comparison with earlier published data.

- L229 Provide more information about the factors that could influence bone strength (incl body mass and BMI, etc.).

- L 249-251 This sentence is unclear, please rewrite the part “participants were relatively high functioning” and specify, which functions are meant.

- After listing the study limitations include information on possible future studies.

- Please add, which exercise programs have been found to be most effective for preventing falls, including their components and recommended duration per week?

- Provide more information on moderate-to-high intensity balance training.

- Please add conclusions that relate to your work hypotheses

Author Response

Responses to the comments from Reviewer #2:

General comments

This is a topical issue and the authors deserve appreciation for their work aimed at elucidating associations between physical functional characteristics, cognitive function, psychological status and fall-related fractures in older population with complaints of pain.

 The description of participants and applied methods is sufficiently clear and the number of participants (389) allows for opening the chosen topics.

The statistical analysis allows the comparison between groups and revealing associations between the measured characteristics. The collected data is presented in three tables and one figure.

In the Discussion the authors unnecessarily repeat the literature review, instead, more comparisons between the collected data and the results from previous publications should be added.

The list of references provides an overview of the related studies, however, the share of the article’s authors in the citations is 16% (n=8 from the total of 50). It is recommended to include more references dating from recent years.

 Response:

Thank you for the review and comments to improve our manuscript. We avoided unnecessary redundancy with respect to the references cited in the discussion section and have included recent relevant references. We also added a few statements to clarify the comparisons between the collected data and the results from previous publications.

Specific comments

The title of the manuscript reflects well the main idea and adequately describes the participants of the conducted research.

 Response:

Thank you for the comment.

Abstract and key words

They are written in the logical manner and include all needed components.

 Response:

Thank you for the comment.

Introduction

Please add the hypothesis of the study.

 Response:

Thank you for the comment. Based on your suggestion, we added the hypothesis of this study in the introduction section as follows:

“We hypothesized that multiple physical, psychological and lifestyle factors would be independently associated with falls in older people with pain.” (page 2, lines 60-62)

Materials and Methods

Participants

Please add the following data:

- previous and/or current sport/fitness activity of participants

- social life (possible influence on the depression level) – family status, partners, children

- employment (elderly persons may continue working)

- previous surgery on the spine or joints (replacements, etc.) that can influence body balance and strength

If the above components were not estimated, include them in the limitations of the study.

Response:

Thank you for the comment. We assessed participants’ physical activity with the Incidental and Planned Exercise Questionnaire (IPEQ) which includes sport and fitness activities in the past three months. This has now been clarified as follows:

“Physical activity was assessed at baseline using the Incidental and Planned Exercise Questionnaire (IPEQ) [29]. The IPEQ provides information regarding the average time spent in hours per week over the past three months in planned physical exercise, in addition to planned walks (e.g. walking in the park) and incidental physical activities (e.g. walking to the store, gardening). The planned physical exercise includes participation in exercise classes, home exercise, other sport/exercises such as swimming, golf, bowls, dancing and jogging. The time spent undertaking all panned and incidental activities is summed to provide an estimate of total physical activity levels (h/week).” (page 3, lines 137-144)

Detailed assessments of participant’s characteristics including social factors, employment and previous surgery on the spine or joints were not assessed in this study. We have included this omission in the limitation section as follows:

“First, we did not assess participant characteristics including social life and employment that can influence depression [62], and previous surgery on the spine or joints that can influence balance and strength [63,64].” (pages 8-9, lines 294-297)

Physical function estimation:

L 107 – add the dimensions of foam pad (width x length x height) and the producer.

 Response: We have now added the detail of the assessments of postural sway as follows:

“Postural sway was measured using a sway meter that recorded displacements of the body at the level of the waist in millimeters while participants stood on a foam rubber pad (70 cm × 60 cm × 15 cm thick) with eyes open for 30 seconds. The device consisted of an inflexible 40-cm-long rod with a vertically mounted pen at its end. The rod was mounted on a 20 cm wide metal plate which was fitted over the participant’s lower back (level of the posterior superior iliac spine) by a firm belt so that the rod extended posteriorly. Fitted firmly, the sway meter offers 1 degree of freedom between the belt and pen as it is free to move in the pitch plane. The pen recorded participant’s postural sway on a sheet of millimeter graph paper, fastened to the top of an adjustable-height table.” (page 3, lines 110-119)

Discussion

Recommendations:

- First present the results of your own study and then comparison with earlier published data.

Response: Thank you for the comment. As we have stated the results of the current study in the first paragraph, we have avoided the repetition of our results in the second and third paragraphs.

- L229 Provide more information about the factors that could influence bone strength (incl body mass and BMI, etc.).

Response: Based on your suggestion, we have provided more information on factors that could influence bone strength as follows:

“In addition to several demographic, anthropometric, lifestyle and body composition factors [44,45], some cross-sectional [46,47] and longitudinal [48] studies have shown that slow gait speed is also significantly associated with reduced bone strength.” (page 8, lines 253-256)

- L 249-251 This sentence is unclear, please rewrite the part “participants were relatively high functioning” and specify, which functions are meant.

Response: Based on your suggestion, we have rewritten this sentence as follows:

“Second, our inclusion criteria meant that participants had relatively high physical function and good mental health so our findings may not be generalizable to frailer people or those with physical or neurological impairments.” (page 9, lines 302-304)

- After listing the study limitations include information on possible future studies.

Response: Based on your suggestion, we stated possible future studies as follows:

“Further prospective cohort studies including detailed participant characteristics, pain assessments and social factors are required to confirm the current findings.” (page 9, lines 316-318)

- Please add, which exercise programs have been found to be most effective for preventing falls, including their components and recommended duration per week?

Response: We have now added effective exercise programs and recommended duration per week for preventing falls as follows:

“Encouragingly, systematic review evidence suggests that exercise programs are effective in addressing poor balance, low mood and inactivity [55,56] as well as in preventing falls in community-dwelling older people [57,58] and that exercise programs that challenge balance and have a high dose (i.e. 3 or more hours per week of prescribed exercise) have the largest effects for fall prevention in older people [57].” (page 8, lines 278-283)

- Provide more information on moderate-to-high intensity balance training.

Response: We have now provided more information on moderate-to-high intensity balance training as follows:

“Thus, it is likely that moderate-high intensity balance training would also be an efficacious fall prevention strategy for those with pain. Moderate-high intensity balance training is defined as exercise while standing and moving the center of mass, narrowing the base of support, and minimizing upper limb support [57]. For example, our systematic review and meta-analysis has demonstrated that both reactive and volitional stepping interventions can prevent falls by approximately 50% in older people in community and institutional settings [61], may be particularly beneficial in this regard.” (page 8, lines 285-291)

- Please add conclusions that relate to your work hypotheses

Response: Based on your suggestion, we have added conclusions that relate to our hypotheses as follows:

“Consistent with our hypothesis that multiple physical, psychological and lifestyle factors would be associated with falls, we found poorer balance, low mood and physical inactivity were independent risk factors for falls in older people with pain.” (page 7, lines 224-226)

Conclusions

Please add the possible future directions of related research.

Response: Based on your suggestion, we added the possible future directions of related research as follows:

“Further prospective cohort studies including detailed participant characteristics, pain assessments and social factors are required to confirm the current findings.” (page 9, lines 316-318)

Reviewer 3 Report

    The authors reported the risks of falls and fall-related fractures in older people with pain. The paper is well written and showed analyzed results clearly. However, the main concern is about the situations of identified risk factors in older people without pain in this population. It might be that identified risks, i.e., poorer balance, low mood, physical inactivity, and the slower walking speed associated with falls/fall-related fractures in older people with pain, may also be significant in older people without pain. Several points are suggested to the authors for considering revisions of the article.

1.     It would be great if the authors could also report findings in older people without pain in this population – if available. Suggest adding a flow chart or reporting on data handling, such as how many participants were removed for analysis in each step, including older people without pain.

2.     Suggest that the authors may highlight the new findings in this study, comparing with publications, which factors or amount (e.g., predictors in Table 2) were more prominent in older people with pain than those without pain.

3.     It is complex relationships among pain, physical inactivity, and psychological factors in the risk of falls or fall-related fractures. The causal relationship is difficult to be defined. Was one-year short for the observation? The authors also mentioned the limitation on lacking pain intensity, duration, number of pain sites, or use of analgesics in this study. Was it possible to use any pain level groups with the lowest as the control to see the risk factors related to falls/fall-related fractures?

4.     Were the cut-off value of the Youden index (3.3 Results) different from other populations? Based on those, were the proportions of participants who experienced falls higher than publications? It might be suitable for some discussion.

5.     Should give a reference for ‘, and a Mini-Mental State Examination score<24’, line 73, 2.1. Participants and study design?

6.     Could the authors list some comorbidities in Methods (2.2.2. Demographics) or Results?

Author Response

Responses to the comments from Reviewer #3:

The authors reported the risks of falls and fall-related fractures in older people with pain. The paper is well written and showed analyzed results clearly. However, the main concern is about the situations of identified risk factors in older people without pain in this population. It might be that identified risks, i.e., poorer balance, low mood, physical inactivity, and the slower walking speed associated with falls/fall-related fractures in older people with pain, may also be significant in older people without pain. Several points are suggested to the authors for considering revisions of the article.

  1. It would be great if the authors could also report findings in older people without pain in this population – if available. Suggest adding a flow chart or reporting on data handling, such as how many participants were removed for analysis in each step, including older people without pain.

Response: Thank you for this suggestion. However, we would prefer to restrict our study to identifying the most salient factors that predict falls and fall-related fractures in those with pain. We feel this is the novel aspect of our work as many previous studies have investigated fall risk factors in older people without pain. We have now stated this in the introduction section as follows:

“However, studies on risk factors for falls have focused on older people primarily without pain [11,12], with little information available on the fall risk profile of older people with pain to inform possible intervention strategies.” (pages 1-2, lines 42-46)

Based on your suggestion, we have also added the study flow chart shown as Figure 1 in the revised version.

“Of 814 participants who took part in the falls and balance component of the Sydney Memory and Ageing Study (waves 1 to 4), 389 (177 men, 212 women, mean age 78.9 years, SD = 4.8) reported having bodily pain, completed the baseline clinical assessments and a 12-month follow-up for falls, and were therefore included in these analyses (Figure 1).” (page 4, lines 163-166)

  1. Suggest that the authors may highlight the new findings in this study, comparing with publications, which factors or amount (e.g., predictors in Table 2) were more prominent in older people with pain than those without pain.

Response: Thank you for this comment. We have now discussed the cut points for the three independent risk factors for predicting falls using the Youden index identified in this study and contrasted these to cut-points found in studies of older people primarily without pain where possible. (page 8, lines 258-274)

  1. It is complex relationships among pain, physical inactivity, and psychological factors in the risk of falls or fall-related fractures. The causal relationship is difficult to be defined. Was one-year short for the observation? The authors also mentioned the limitation on lacking pain intensity, duration, number of pain sites, or use of analgesics in this study. Was it possible to use any pain level groups with the lowest as the control to see the risk factors related to falls/fall-related fractures?

Response: Indeed, the causal relationship between pain and physical, cognitive, psychological and lifestyle risk factors for falls are difficult to tease out because many these factors are chronic. We have now stated this in the limitation section as follows:

“Finally, the causal relationship between pain and physical, cognitive, psychological and lifestyle risk factors for falls is difficult to tease out because many of these factors are chronic and some may have existed before the commencement of this prospective study. Further research detailed participant characteristics, pain assessments and social factors and with a longer-term follow-up is required to confirm the current findings.” (page 9, lines 305-310)

We did not include a detailed assessment of pain intensity, so we are not able to contrast fall and fracture rates with respect to different pain levels. As indicated above we have acknowledged the omission of a detailed pain assessment as a study limitation.

  1. Were the cut-off value of the Youden index (3.3 Results) different from other populations? Based on those, were the proportions of participants who experienced falls higher than publications? It might be suitable for some discussion.

Response: Based on your suggestion, we added some discussion related to the cut points of the Youden index as follows:

“The cut points for the three independent risk factors for predicting falls using the Youden index requires discussion. The best cut point for postural sway while standing on a foam rubber pad was 200 mm. This value is higher than that reported in studies of community-dwelling older people where the best cut points for predicting falls have ranged from 105 to 168 mm [49-51]. It is likely, the higher cut-point found here reflects overall worse balance in our sample with pain. [14]. Our finding that that the relatively low score of 3 on the GDS-15 best discriminated between fallers and non-fallers suggests that even mild depressive symptoms is a risk factor for falls in older people with pain [52]. Finally, to our knowledge, this is the first study to investigate a cut-point of combined incidental and planned physical activities to predict falls in older people with pain. Our cut-point indicating less than 34 hours/week of total physical activity is predictive of future falls is consistent with previous prospective cohort studies that have reported physical inactivity increases fall risk [53,54]. However, there is also evidence of non-linear (e.g. u-shaped) associations between falls and physical activity [42], with greater fall risk in inactive/impaired older people as well as active older people with high environmental exposure. Thus, cut points may vary depending on sample characteristics such as frailty and disability.” (page 8, lines 258-274)

  1. Should give a reference for ‘, and a Mini-Mental State Examination score<24’, line 73, 2.1. Participants and study design?

Response: We have now provided the following reference for “a Mini-Mental State Examination score < 24 [19]”.

  1. Tombaugh, T.N.; McIntyre, N.J. The mini-mental state examination: a comprehensive review. Journal of the American Geriatrics Society 1992, 40, 922-935, doi:10.1111/j.1532-5415.1992.tb01992.x.

  1. Could the authors list some comorbidities in Methods (2.2.2. Demographics) or Results?

Response: We have now provided the following description on comorbidities in the footnote for Table 1.

“Comorbidity indicates the presence of 2 or more of the follow conditions: stroke, Parkinson disease, diabetes mellitus, chronic obstructive pulmonary disease and heart disease.” Footnote for Table 1

Round 2

Reviewer 3 Report

Thank you for taking the effort in revising the interesting article. I have no comments.